# Localization of Epigenetic Markers in *Leishmania* Chromatin

**DOI:** 10.3390/pathogens11080930

**Published:** 2022-08-18

**Authors:** Jacquelyn R. McDonald, Bryan C. Jensen, Aakash Sur, Iris L. K. Wong, Stephen M. Beverley, Peter J. Myler

**Affiliations:** 1Center for Global Infectious Disease Research, Seattle Children’s Research Institute, Seattle, WA 98109, USA; 2Department of Biomedical Informatics and Medical Education, University of Washington, Seattle, WA 98195, USA; 3Department of Molecular Microbiology, Washington University School of Medicine, St. Louis, MO 63110, USA; 4Department of Pediatrics, University of Washington, Seattle, WA 98195, USA

**Keywords:** *Leishmania*, epigenome, histone variants, post translation modifications

## Abstract

Eukaryotes use histone variants and post-translation modifications (PTMs), as well as DNA base modifications, to regulate DNA replication/repair, chromosome condensation, and gene expression. Despite the unusual organization of their protein-coding genes into large polycistronic transcription units (PTUs), trypanosomatid parasites also employ a “histone code” to control these processes, but the details of this epigenetic code are poorly understood. Here, we present the results of experiments designed to elucidate the distribution of histone variants and PTMs over the chromatin landscape of *Leishmania tarentolae*. These experiments show that two histone variants (H2A.Z and H2B.V) and three histone H3 PTMs (H3K4me3, H3K16ac, and H3K76me3) are enriched at transcription start sites (TSSs); while a histone variant (H3.V) and the trypanosomatid-specific hyper-modified DNA base J are located at transcription termination sites (TTSs). Reduced nucleosome density was observed at all TTSs and TSSs for RNA genes transcribed by RNA polymerases I (RNAPI) or RNAPIII; as well as (to a lesser extent) at TSSs for the PTUs transcribed by RNAPII. Several PTMs (H3K4me3, H3K16ac H3K20me2 and H3K36me3) and base J were enriched at centromeres, while H3K50ac was specifically associated with the periphery of these centromeric sequences. These findings significantly expand our knowledge of the epigenetic markers associated with transcription, DNA replication and/or chromosome segregation in these early diverging eukaryotes and will hopefully lay the groundwork for future studies to elucidate how they control these fundamental processes.

## 1. Introduction

Trypanosomatid parasites cause widespread suffering and death in humans and animals, resulting in a substantial economic burden in many countries around the world [1]. *Leishmania* are transmitted by sand flies, in which they proliferate as procyclic promastigotes and differentiate into non-dividing metacyclic promastigotes. After transmission to the vertebrate host, metacyclics infect macrophages and differentiate into amastigotes, which grow and divide within the phagolysosome and ultimately lyse the host cell to serially infect other macrophages [2]. Infection can lead to several different disease outcomes, depending on the species and several host factors [3], ranging from asymptomatic or self-limiting cutaneous infections to debilitating diffuse cutaneous leishmaniasis, disfiguring mucocutaneous leishmaniasis, and lethal visceral disease.

The nuclear genome of all eukaryotes (including *Leishmania* and other trypanosomatids) is packaged into chromatin by interaction with four core histone (H2A, H2B, H3 and H4) homodimers to form heteromeric nucleosomes that each have ~150 bp of DNA wrapped around them. The nucleosomes are joined by a linker histone (H1) to form higher-order structures, compacting the DNA and protecting it from damage. DNA base modification and post-translational modifications (PTMs) on the histones (and their variants) are used to control diverse biological processes, including DNA replication/repair, chromosome condensation, and temporal regulation of gene expression. Lysine and arginine residues in N-terminal “tails” of the core histones protruding from the nucleosome are covalently modified by methylation, acetylation, or ubiquitination; while serine or threonine residues are phosphorylated. These PTMs can change the local nucleosome density, thereby controlling the rate of transcription initiation, elongation, and termination. The number of PTMs is large, and our understanding of the functional significance of this epigenetic “histone code” is still in its infancy, even for humans and model organisms [4].

In most eukaryotes, each gene normally has its own transcription start site (TSS) and termination site (TTS), allowing for regulation of mRNA production on an individual basis. In contrast, trypanosomatid genomes are organized into polycistronic transcription units (PTUs) containing tens-to-hundreds of protein-coding genes on the same strand of DNA [5]. The primary transcript from each PTU is processed co-transcriptionally by trans-splicing and polyadenylation, resulting in addition of a capped 39-nucleotide spliced leader (SL) sequence to the 5′-end and a poly(A) tail to the 3′ end of each mature mRNA [6]. Consequently, trypanosomatid genomes contain far fewer transcription start sites (TSSs) and transcription termination sites (TTSs) than other eukaryotes and they rely almost exclusively on post-transcriptional mechanisms (RNA processing, stability and translation) to regulate differential gene expression, with the 5′ and 3′ untranslated regions (UTRs) being the primary drivers of differential mRNA abundance [7]. However, despite the (mostly) constitutive transcription of their protein-coding genes, trypanosomatids still utilize histone variants and PTMs to mark the boundaries of their PTUs [8,9,10,11]. In addition, a trypanosomatid-specific hyper-modified DNA base (β-D-glucopyranosyloxymethyl-uracil (J for short) [12], which occurs mostly at telomeres [13], is also found at TTSs [14], as well as some divergent TSSs [15,16]. In *Leishmania*, loss of J leads to a substantial increase in read-through transcription at TTSs [14]; but in *T. brucei* it appears to act more synergistically with H3.V to control transcription termination and expression of non-coding siRNAs [17]. In contrast, reduction of J levels in *T. cruzi* is accompanied by a significant increase in transcription of virulence-associated genes (such as trans-sialidases, gp85, and amastins) located between PTUs, suggesting that J may also have a role in regulating transcription initiation in some species [18].

While trypanosomatids have orthologues for all the canonical histones, they are quite divergent from those in other organisms [19,20,21,22]. *Leishmania* H1 contains a single domain that corresponds to the lysine-rich C-terminus of H1 in higher eukaryotes, but few of the residues with PTMs in human H1 are conserved [23]. *Leishmania* H2A and H2A.Z both appear to represent a sister branch to the H2A.Z variant of higher eukaryotes. Most of the lysines that have PTMs in human H2A/H2A.Z are conserved, but *Leishmania* H2A.Z contains a larger N-terminal extension with more PTMs than in other organisms [24]. *Leishmania* H2B has a shorter N-terminal tail than in mammals, with only two of the modified lysines being conserved (although a third may be conserved in *T. brucei*). However, H2B.V in *Leishmania* (along with other trypanosomatids) has a longer tail containing additional lysines, which may be the functional equivalent of H2B residues that are acetylated in actively transcribed human chromatin [25]. H3 and H3.V from trypanosomatids are quite divergent from those in other eukaryotes and both appear to branch with the centromeric H3 variant (cenH3 or CENP-A) in other taxa. Trypanosomatid H3 has a shorter N-terminal tail than in most other organisms, although most of the lysines with PTMs in human (and/or yeast) H3 are conserved. The H3 variant (H3.V) of both *Leishmania* and *Trypanosoma* contains a longer N-terminal tail (with several additional lysine residues), although there are differences between the two species [26]. H4 is the most conserved of the trypanosomatid histones, with almost all residues with PTMs in human H4 being conserved. *T. brucei* has an H4 variant (H4.V), but *Leishmania* appears to lack an obvious orthologue [27], although there are small differences between some copies of H4.

Eukaryotes have evolved a panoply of “readers” and “writers” that recruit chromatin modification complexes and alter local nucleosome density, respectively, thereby controlling its accessibility to RNA polymerase and transcription factor complexes [4,28]. The epigenetic “histone code” is complicated and incompletely understood, even for humans and model organisms. Previously published reports have identified >150 different PTMs on the core histones and variants of *T. brucei* [11] and ~200 for *T. cruzi* [22]. A more recent publication [24] identified 58 different PTMs enriched at TSSs in *T. brucei*, including trimethylation of H3K4, H3K10 and H3K11, acetylation of H3K23, and acetylation of several lysines on H2A.Z, H2B.V and H4. Interestingly, at least two different histone acetyltransferases (HATs) are involved in acetylation of *T. brucei* histones; HAT1 is primarily responsible for acetylation of H2A.Z and H2B.V, while HAT2 acetylates H4 [24]. There have been fewer studies in *Leishmania*, but H3K9ac (the equivalent of H3K10ac in *T. brucei*) has been shown enriched at TSSs [8,29,30,31] and H3.V at TTSs [27] in *L. major*; while H4K4ac, H4K10ac and H4K14ac are enriched at TSSs in *L. donovani* [8,29,30,31]. Here, we describe experiments that utilize a recently developed improvement of chromatin immunoprecipitation (ChIP-seq) technology [32] to significantly expand our knowledge of the chromatin-wide distribution of histone variants and PTMs in *L. tarentolae.*

## 2. Results

To map the location of histone variants and PTMs across the chromatin landscape of *L. tarentolae*, we employed Cleavage Under Targets and Tagmentation (CUT & Tag) [33], in which a target within chromatin is bound in situ by a specific antibody, thereby directing a protein A-Tn5 transposase fusion protein (pA-TN5) to this location. Addition of bar-coded oligonucleotides and activation of the transposase cleaves the adjacent DNA and creates “tagged” ends, allowing generation of fragment libraries with high resolution and exceptionally low background. In addition, we repeated our previous J-IP-seq experiments [14] that showed J is enriched at telomeres, TTS and some TSSs at divergent strand-switch regions. The Illumina sequencing reads obtained from these experiments were mapped to the *L. tarentolae* genome using a short-read aligner and the read density plotted for each of the 36 chromosomes in the genome. Figure 1 shows the results obtained for chromosome 25, which contains two examples of divergent TSSs, two convergent TTSs, and two telomeric TTSs; as well as a unidirectional TTS/TSS (i.e., a TTS followed by a TSS on the same strand), which is located at the centromere. This distribution of PTUs (and chromatin markers) on chr25 is representative of the ~192 PTUs found on the 36 *Leishmania* chromosomes, although some minor differences were occasionally seen, as outlined below.

H2A and H2B (to a lesser extent) were depleted at all TSSs, irrespective of whether they were bi-directional, unidirectional or telomeric (Figure 1B), consistent with their replacement by H2A.Z and H2B.V at these locations (Figure 1C). As expected, H3.V was enriched at TTSs and telomeres (Figure 1C). Interestingly, H3 showed lower coverage at TTSs and TSSs (Figure 1B), possibly reflecting lower chromatin density at these locations (see below). We also found that three H3 PTMs (H3K4me3, H3K16ac and H3K76me3) were enriched at TSSs (Figure 1D).

The centromeres of all 36 *L. tarentolae* chromosomes were identified by chromatin confirmation capture followed by next generation sequencing (Hi-C) and confirmed by synteny with *L. major* [34]. The centromeres (Appendix A) are located between divergent TSSs (14/36), unidirectional TTS/TSSs (16/36), or convergent TTSs (6/36) and are sometimes (9/36) flanked by RNA genes. Figure 2 shows examples of the chromatin markers found at each of the six different configurations for centromeres. H3K4me3 and H3K16ac appeared to be enriched at centromeres, even those not associated with TSSs. In contrast, H3K20me2, H3K36me3 and H3K50ac were found only at centromeres. Interestingly, while H3K4me3, H3K20me2 and H3K36me3 (and perhaps H3K16ac) were distributed throughout the entire centromeric region, H3K50ac showed peaks mostly at the centromere boundaries.

In agreement with our previous results [14], base J is found at all TTSs (except for the second convergent TTS on chr28) and telomeres, as well as centromeres (Figure 1E and Figure 2E). Significantly, the latter also account for most of the instances where J was found between bi-directional TSSs (e.g., chr06 and chr02, Figure 2E). Finally, we performed an Assay for Transposase-Accessible Chromatin using sequencing (ATAC-seq) experiment, which identifies regions with “open” (i.e., more loosely packed) chromatin [35]. Peaks of read density were observed at TTSs (Figure 1F and Figure 2F), reflecting the reduced nucleosome concentration previously observed at these sites [36]; but we also saw (less-intense) peaks at most TSSs, suggesting that chromatin may be loosely packed at these locations, as well. However, the ATAC-seq data showed no increase in signal within the central centromeric sequence (Figure 2F), suggesting that chromatin density at centromeres was similar to that found within the PTUs, although the read density of histones (and their variants) from the CUT & Tag experiments was generally lower at centromeric regions (Figure 2B,C). As expected, read density from ATAC-seq was highest wherever RNA genes were transcribed by RNAP I (rRNA locus on Chr27) or RNAP III (tRNAs and 5S rRNAs on several chromosomes), consistent with the high transcription rate at these loci.

## 3. Discussion

Eukaryotes use histone variants to modify nucleosome structure and facilitate transcription initiation, DNA replication/repair and chromosome segregation at specific loci. The trypanosomatids are no exception and our results confirm that the distribution of histone variants in *Leishmania* is similar to that previously reported for *T. brucei* [9,11], with H2A.Z and H2B.V marking the TSS at the beginning of each PTU of protein-coding genes transcribed by RNAPII. A large number of PTMs have been found on histones present at TSSs in *T. brucei* [24,37], and we confirmed that at least three of these (H3K4me3, H3K16ac and H3K76me3) are also present at *Leishmania* TSSs (Table 1). These markers, which are typically associated with transcriptional activation in other eukaryotes [38,39], were also found upstream of RNA genes transcribed by RNAPI (rRNA) and RNAPIII (5S rRNA, snRNA, tRNA or SRP RNA).

In *T. brucei*, the histone variants H3.V and H4.V are associated with the TTS at the end of each PTU, along with base J [9]. As expected, our results confirm that H3.V and base J are similarly localized in *Leishmania* (Table 1). However, *Leishmania* appears to lack an obvious orthologue of the H4 variant (H4V) found in *T. brucei* [27], providing further evidence for differences between the relative importance of these chromatin markers for transcription termination in *Leishmania* and *Trypanosoma* [26].

Previous studies using *L. major* have shown that TTSs have a low occupancy of nucleosomes [36], suggesting that the presence H3.V and/or base J results in chromatin remodeling at these sites. The results of our ATAC-seq experiments (Table 1) are consistent with less densely packed chromatin at these loci, perhaps facilitating binding of protein complex(es) that mediate termination. We also found open chromatin at TSSs of RNA genes transcribed RNAPI and III, probably to facilitate increased access by these RNA polymerase initiation complexes. We also found an increase (albeit less pronounced) in ATAC-seq read density at the TTSs of PTUs transcribed by RNAPII, suggesting that chromatin at these sites is also somewhat open, as seen in for actively transcribed genes in other eukaryotes.

In other eukaryotes, a histone H3 variant (CENP-A) is used to generate more rigid nucleosomes at centromeres [40], although some H3 PTMs (e.g., H3K4me2) are also important in kinetochore formation and chromosome segregation [41]. To our knowledge, our study is the first to characterize histone markers associated with centromeres in trypanosomatids, which do not have a clear CENP-A orthologue (although their H3 appears more closely related to cenH3/CENP-A than to the “canonical” H3). Our results found that several PTMs are associated with centromeres in *L. tarentolae* (Table 1). Two (H3K4me3 and H3K16ac) are shared with TSSs, but three (H3K20me2, H3K36me3 and H3K50ac) are found only at centromeres. Base J is also enriched at centromeres (even those not flanked by obvious TTSs). Interestingly, H3K50ac appears to be located at the periphery of the centromeric sequence, while the other markers are found throughout the centromere. The density of histones (H2.A, H2.B, H3 and H4) and their variants (H2A.Z, H2B.V and H3.V) appears to be reduced within the central centromeric region. Surprisingly, this did not appear to be accompanied by an increase in ATAC-seq read density, although the AT-rich nature of most centromeres may have confounded this analysis. These results suggest that relationship between chromatin modification and centromeric function in *Leishmania* is significantly different from that in other, more well-studied, eukaryotes. It remains to be seen whether the situation is similar in *T. brucei*.

## 4. Materials and Methods

### 4.1. Parasite Cell Culture

A clone (M2) of *L. tarentolae* Parrot TarII [42] was grown in SDM-79 supplemented with 10% fetal calf serum [43]. In general, cultures were grown to mid-log phase and harvested at a density of ~10^7^ cells/mL.

### 4.2. Assembly and Annotation of the Reference Genome

Sequencing libraries were prepared from genomic DNA of *L. tarentolae* Parrot TarII using the BluePippin^TM^ System with a 7.5 kb size-cut (https://pacbio.gs.washington.edu/ (accessed on 30 July 2022)). The library was sequenced using P2/C2 chemistry on 14 SMRTCells to generate 2,104,088 reads with an average read length of 3241 nt. Filtering for duplicates reduced this to 913,321 reads (average length = 5.408 nt), which were assembled using HGAP v3.0 [44], resulting in 177 contigs, of which 70 likely represented an alternative haplotype and were discarded, along with 2 that represented the maxicircle. The remaining contigs were concatenated into 36 chromosomes by comparison to exiting *L. major* and *L. tarentolae* reference genomes using ABACAS [45]. This sequence was error-corrected with iCORN [46] after alignment to several Illumina whole genome sequencing libraries, followed by PBJelly [47] to fill sequence gaps and manual correction to correct obvious remaining errors. The resultant 32,192,075-bp sequence was annotated using RATT [48], SNAP [49], Augustus [50], Aragorn [51] and Infernal [52], followed with by limited manual refinement to provide an updated version (LtaP_2016) of the *L. tarentolae* reference genome(see Appendix A). The CDSs were manually curated into 192 PTUs, based largely on the location of the peaks of base J/H3V and H2A.Z/H2B.V. shows the final size of each chromosome, along with the number of sequence gaps, different gene types and PTUs.

Hi-C analysis [10] was used to localize the centromeres of all 36 *L. tarentolae* chromosomes (see Appendix A). Briefly, Hi-C libraries were prepared by Phase Genomics, Seattle USA using their Proximo Hi-C kit and sequenced with Illumina 80-bp paired end technology. The resultant 59,964,481 read pairs were aligned against the LtaP_2016 reference genome using Bowtie2 [11] and the results viewed in a HiGlass browser [12]. We found that each chromosome contained a single off-diagonal region of high-proximity with every other chromosome. These regions were syntenic with the centromeres previously identified in *L. major* [13]. The exact boundaries of the centromeric sequences were refined by manual inspection of RNA-seq data to choose those regions with the lowest coverage.

### 4.3. CUT & Tag

A previously published protocol [33] was used to determine the chromatin localization of canonical and variant histones, as well as selected PTMs. Briefly, Concanavalin beads were added to ~10^7^ cells that had been permeabilized with digitonin, followed by addition of 1 μL of specific antibody and overnight incubation at 4 °C with rotation. Antibodies prepared against *L. major* histones, histone variants and PTMs were provided by Iris Wong and Stephen Beverley. Antisera against unmodified histone and their variants were prepared against recombinant protein in a manner similar to that described from H3.V [27]. Antisera to *L. major* histone modifications (K4me, K16ac, K20me2, K36me3 and K50ac) were prepared by synthesis of ~11 amino acid peptides with the modification at the central position, conjugation to carrier protein, and immunization of rabbits. Antisera were evaluated for specificity by Western blotting against modified or unmodified peptides (IJK Wong and SM Beverley, manuscript in preparation). Antibodies against *T. brucei* H3 and H3K76me3 were supplied by Christian Janzen [53]. After overnight incubation, the cells were washed before the secondary antibody (guinea pig anti-rabbit IgG) was added and incubated for 30 min at room temperature. After the cells were washed to remove unbound antibody, protein A-Tn5 complexed with oligonucleotide adapters (kindly supplied by Steven Henikoff) was added and the cells incubated for 60 min at room temperature. Tagmented DNA was subsequently purified and converted into Illumina sequencing libraries.

### 4.4. J-IP-Seq

To localize base J within the *L. tarentolae* genome, we used a modification of the previously described protocol [14]. DNA (20 μg) was fragmented to ~200-bp fragments using a Covaris S2 and purified using Ampure XP beads (Beckman Coulter, Indianapolis, IN, USA). The DNA concentration was determined using a Qubit HS (ThermoFisher, Waltham, MA, USA) and 10 μg diluted to 500 μL with binding buffer (PBS with 0.02% Tween-20). Ten μL of antibody against base J [54] was diluted to 200 μL and added to 50 μL of Protein A Dynabeads (ThermoFisher, Bothell, WA, USA) before being incubated with rotation for 40 min at room temperature. The supernatant was removed, and the beads washed with 400 μL buffer. The sheared DNA was then added to the beads and incubated for a further 50 min, before the supernatant was removed and beads washed three times with wash buffer (10 mM Tris pH 8.0, 8 mM EDTA, 85 mM NaCl, 0.05% Tween-20). To reduce background reads from unbound DNA stuck to the sides of the tube, beads were then resuspended in 200 μL of wash buffer and transferred to a fresh tube. The tube was then placed on a magnet and the supernatant removed before the beads were resuspended in 250 μL elution buffer (10 mM Tris pH 8.0, 1 mM EDTA, 0.1% SDS) and 50 ug of Proteinase K, followed by incubation at 56 °C for 30 min in a thermomixer. The eluted DNA was purified by sequential phenol: chloroform: isoamyl alcohol and chloroform: isoamyl alcohol extractions before precipitation and resuspension in TE. Libraries were prepared from 1 ng of DNA using the NEBnext ultra II Library preparation kit (New England Biolabs, Ipswich, MA, USA) with 10 amplification cycles.

### 4.5. ATAC-Seq

Libraries were prepared as described previously [55], using 10^7^ cells per sample.

### 4.6. Sequencing and Read Mapping

The Illumina libraries described above were sequenced by Novogene Corporation Inc., Sacramento, CA, USA, using their NovaSeq PE150-30G-WOBI protocol. Reads were aligned against the *L. tarentolae* reference genome (LtaP_2016) described above, using Bowtie2 [56] with the “end-to-end alignment” and “low sensitivity/fastest” options within Geneious Prime (2022.1.1). The number of reads aligned for each library is shown in Appendix A.

## Figures and Tables

**Figure 1 pathogens-11-00930-f001:**
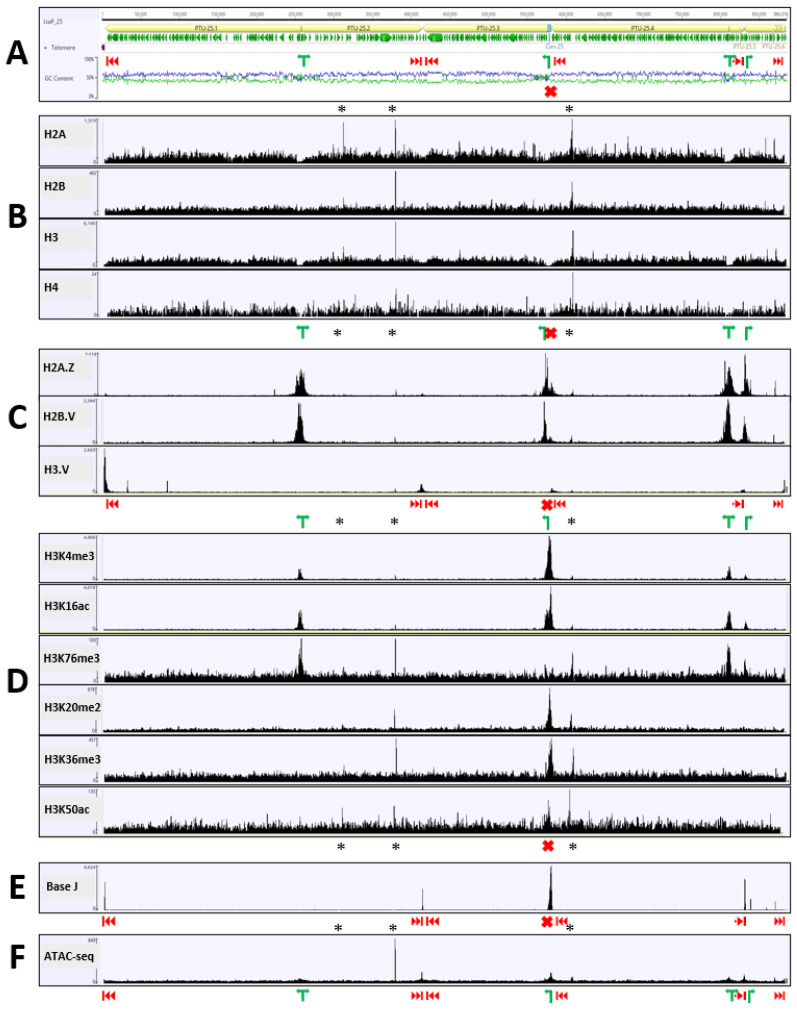
Distribution of epigenetic markers across a representative chromosome. The top panel (**A**) Scheme 25. which are organized into six PTUs (pale yellow). Telomeric hexamer repeats (THRs) are denoted by the purple arrowheads at the left end (the reference sequence ends short of the THRs at the right end of chr25). Transcription start sites (TSSs) are indicated by green arrows, transcription termination sites (TTSs) by vertical bars following red arrowheads, while the centromere (shown by a blue box in the PTU map) is indicated by red X. Base composition in 100-bp sliding windows across the chromosomes is indicated by blue (%GC) and green (%AT) lines. The coverage maps in the lower panels (**B**–**F**) represent the number of reads mapped by Bowtie2 mapping at each nucleotide. Sharp peaks (indicated by asterisks) present in all panels (including controls) likely represent short repetitive sequences, rather than actual peaks of histone variants or PTMs.

**Figure 2 pathogens-11-00930-f002:**
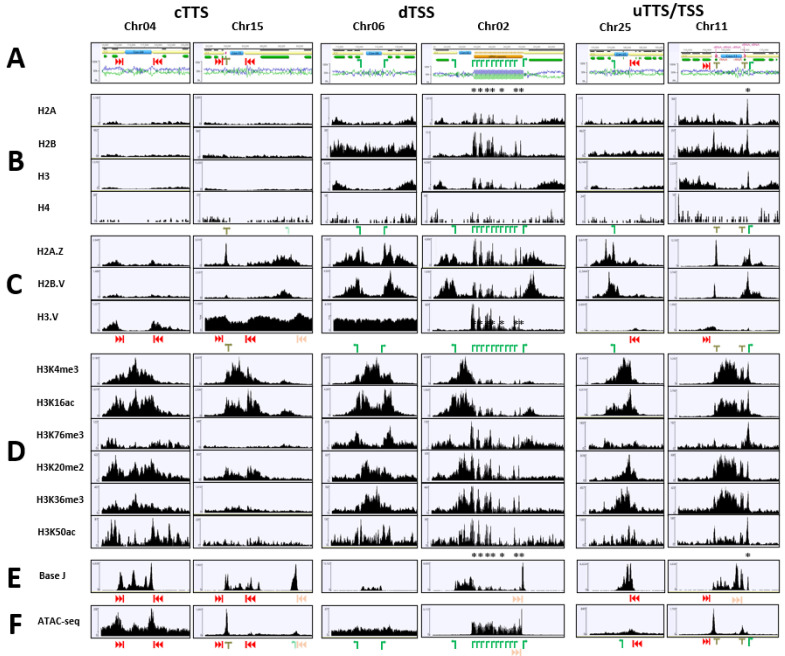
Distribution of epigenetic markers at centromeres. The top panel (**A**) shows the location of protein-coding and RNA genes surrounding the centromeric region of six different *L. tarentolae* chromosomes, with two examples each representative of centromeres between convergent TTSs, divergent TSSs, or unidirectional TTS/TSSs. The right-hand example of each pair contains RNA genes flanking one or both sides of the centromeric sequence, which is indicated by a blue box between the adjacent PTUs (yellow arrows). Transcription start sites (TSSs) are indicated by a green arrow and the transcription termination sites (TTSs) are denoted by vertical bars following red arrowheads (lighter shades indicate TSSs and TTSs for RNA genes). The base composition in 100-bp sliding windows across the chromosomes is shown by blue (%GC) and green (%AT) lines. The coverage maps in the lower panels (**B**–**F**) represent the number of reads mapped by Bowtie2 mapping at each nucleotide. A log_2_ scale was used for the coverage map of H3.V on chr15 and chr06, while all the other maps used a linear scale.

**Table 1 pathogens-11-00930-t001:** Distribution of histone variants and PTMs across the *Leishmania* chromatin landscape.

Location	Marker (s)	Chromatin State
Telomeres	H3.V, J	Unclear
TSSs	H2A.Z, H2B.V, H3K4me3, H3K16ac, H3K76me3	Open
PTU body	H2A, H2B, H3 and H4	Normal
TTSs	H3.V, J	Open
Centromere	H3K4me3, H3K16ac, H3K20me2, H3K36me3, (H3K50ac) *	Normal

* H3K50ac was found only at centromere boundaries.

## Data Availability

All CUT & Tag, J-IPseq and ATC-seq sequencing libraries have been submitted to the GenBank Short Read Archive under BioProject PRJNA861905.

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
