# Peer review of "Localization of Epigenetic Markers in Leishmania Chromatin"

_pathogens, 2022, doi:10.3390/pathogens11080930_

Round 1

Reviewer 1 Report

This is an exceptional article (of the kind that might be found in a leading, broad-interest journal). The study provides a complete picture of the Leishmania chromosomal architecture. Thus, the authors present the results of different experiments designed to elucidate the distribution of histone variants (H2A.Z, H2B.V and H3.V) over the chromatin landscape of Leishmania tarentolae. Also, they documented regions in which several histone H3 PTMs accumulate differently. Also, the distribution of base J was analyzed, and the degree of chromatin compaction. In sum, as the authors indicate, these findings significantly expand our knowledge of the epigenetic markers associated with transcription, DNA replication and chromosome segregation in Leishmania.

            The manuscript is well-written, the results are solid and the interpretations are convincing. I have a few comments to share with the authors, who may consider incorporating them into the manuscript during its revision.

Specific points

1. Lines 83-85.- The authors should clarify which organism (either Leishmania or T. brucei) they are referring to. Similar confusion may arise in the following sentences, as the quoted references are for studies done in T. brucei.

2. Line 90.- Something is missing: H3 and H3.V of are

3. Lines 241-242.- Please, indicate cell density at mid-log phase.

4. Line 249.- Please, specify which peptides were used for rabbit immunizations.

5. Lines 126 and 285.- The authors indicate that an in-house genome version was used. It is relevant that this sequence is publically available.

6. Lines 141-143.- It is indicated how centromeres were mapped; however, a brief description of the HiC methodology foundations would be appreciated in order to better understand how the chromosomal centromeres are defined.

7. Line 136, figure 1C and materials and methods (M&M) section.- The origin of anti-H2A.Z and H2B.V antibodies should be clarified. Although in the M&M section it is stated that the production of those antibodies is described in reference 27, only the description of anti-H3V antibody production was found by this reviewer in the quoted article.

8. Line 146. What kind of RNA genes?

Reviewer 2 Report

This paper represents some initial work on determining the histone code of Leishmania. The paper is well written and the results appear to be relatively strong. It will be of interest to those studying epigenetics and the histone code.

On line 134 authors indicate that H2A and H2B appear depleted. H2B does not really look depleted when one looks at the figures. Or at least it is not as depleted as H2A, H3 and H4 in those regions. However, this could be a resolution issue or it might not be important in that the presence of H2A.Z and H2B.V in those regions does appear solid. 

Also the figures in the printed form are quite small and difficult to see the details. And when one zooms in the resolution is rather bad. Putting high-resolution graphics as an appendix may be helpful. 

Didn't see the supplementary Table A1. I assume Table B1 in Appendix A has been mislabelled. The description in Appendix A is written in lab jargon that is essentially incomprehensible and there is no explanation of what the table is showing. 
